# Molecular Survey of *Hepatozoon canis* Infection in Domestic Dogs from Sardinia, Italy

**DOI:** 10.3390/vetsci10110640

**Published:** 2023-10-31

**Authors:** Valentina Chisu, Laura Giua, Piera Bianco, Giovanna Masala, Sara Sechi, Raffaella Cocco, Ivana Piredda

**Affiliations:** 1Istituto Zooprofilattico Sperimentale “G. Pegreffi” della Sardegna, Via Duca degli Abruzzi 8, 07100 Sassari, Italy; laura.giua@izs-sardegna.it (L.G.); piera.bianco@izs-sardegna.it (P.B.); giovanna.masala@izs-sardegna.it (G.M.); 2Teaching Veterinary Hospital, University of Sassari, Via Vienna 2, 07100 Sassari, Italy; sarasechilavoro@tiscali.it (S.S.); rafco@uniss.it (R.C.)

**Keywords:** dog, hemoparasites, *Hepatozoon*, tick-borne pathogens

## Abstract

**Simple Summary:**

This study was planned to investigate the presence of *Hepatozoon* species in 51 domestic dogs in Sardinia, Italy. The parasites infecting domestic dogs underwent genetic and phylogenetic analyses. The results of PCR tests based on the 18S rRNA gene illustrated that 13.5% (9/51) of the samples were positive for *Hepatozoon* spp. DNA. From the obtained positive samples, the nucleotide sequences of the 18S rRNA gene were also determined. In the genetic and phylogenetic analyses, all nucleotide sequences exhibited a close similarity to *Hepatozoon canis*. The widespread presence of this protozoan species among domestic dogs in Sardinia paves the way for a more accurate description of the epidemiology of these protozoans in the study area.

**Abstract:**

Tick-borne protozoans of the genus *Hepatozoon* have been associated with infections of domestic and wild animals over the world. The occurrence of these apicomplexan agents in Sardinia has been poorly explored so far. In this study, the occurrence of *Hepatozoon* spp. has been investigated in domestic dogs from nine cities of Sardinia, Italy. Blood samples from each dog were collected and tested molecularly for the presence of *Hepatozoon* and *Babesia/Theileria* DNAs. Out of fifty-one dogs, nine were positive for *Hepatozoon* species based on the molecular detection of the parasite in blood samples. The phylogenetic relationships of strains detected here were also established. The PCR for amplification of the 18S rRNA fragment gene of *Babesia*/*Theileria* spp. did not give amplicons in any of the analyzed samples. Our results report the first molecular confirmation of *Hepatozoon canis* in Sardinian pet dogs and contribute to better understand the presence of these protozoans on the island. This study highlights the importance of recognizing and predicting the risk levels for the canine population, thus increasing the development of specific control measures. Also, since the distribution of hepatozoonosis is closely related to that of the definitive tick host, *Rhipicephalus sanguineus*, more accurate studies on *Rhipicephalus* ticks will be needed due to increasing the epidemiological knowledge of *Hepatozoon* species on the island.

## 1. Introduction

Species belonging to *Hepatozoon* are protozoan of the phylum Apicomplexa, family Hepatozoidae of the suborder Adeleorina, and order Piroplasmida [1]. The most representative morphological characteristic of the apicomplexan vector-borne blood parasites is the presence of the apical complex in the cellular conical pole. This last structure is required to penetrate into the host cells, and it is composed of one or two polar rings, a conoid, rhoptries, microtubules subpellicular, and micronemes, which are modified secretory vesicles. All these components have a pivotal role in the biology of the parasite [2]. 

The *Hepatozoon* genus comprises about 340 species that infect and cause diseases of veterinary importance in a broad host range of vertebrate hosts such as mammals, reptiles, birds, fishes, and anurans [3,4,5,6]. Hematophagous invertebrates act as hosts and/or definitive vectors since they are also capable of transmitting these apicomplexan parasites [7]. The different protozoan species included in the genus *Hepatozoon* show different preferential host and cell tropism. Many *Hepatozoon* spp. enter the bloodstream and infect the red blood cells of the affected vertebrates (especially snakes) [7]. However, other species (*H. americanum* and *H. canis*) invade leukocytes with acarines such as ticks, mites, and fleas, serving as definitive hosts or vectors of the parasite [8].

The biological cycle of *Hepatozoon* is complex and requires two or more hosts: (a) the intermediate vertebrate hosts in which the asexual development (merogony or schizogony) and the subsequent formation of gamonts (gamontogonic phase) take place; (b) the invertebrate vectors (the definitive hosts) in which the gamogonic (fertilization) and sporogonic (oocyst formation) phases occur [9].

Specifically, vertebrates become infected after ingestion of arthropods infected with mature oocysts. The sporozoites are then released in the gastrointestinal tract of the vertebrate host, and after the gut wall is broken, they reach the target tissues hematogenously or via lymphatic circulation. Within these tissues, the merogonic phase occurs with meronts formation. The macromerozoites form secondary meronts, continuing the asexual cycle of merogony, while the micromerizoites invade monocytes and neutrophils to form gamonts. In parasitemic mammals, it is also possible to detect the presence of gamonts in circulating leukocytes. They invade new cells and penetrate the cytoplasm of leukocytes (monocytes and neutrophils). The cycle is completed when the hematophagous host ingests the gamonts via the blood meal on the intermediate host vertebrate. Sporogony occurs in the hemocoel after the gametogenesis within the tick gut [10,11]. 

Other routes of infection have also been described in different *Hepatozoon* species. These include the transmission by ingestion of monozoic cysts from paratenic hosts during predation of infected animals. It happens for some *Hepatozoon* spp., such as *H. americanum*, *H. sipedon, H. caimani*, and *H. ayorgbor,* that infect canids, reptiles, caiman crocodiles, and snakes, respectively [12]. 

However, this last mechanism has not been described for *H. canis*, in which conversely, vertical transmission has been reported [13]. In addition, intrauterine transmission has also been recently demonstrated, and *H. canis* gamonts have been observed in peripheral blood smears of puppies in Japan [11]. Also, a recent study reported the positivity for *H. canis* in the spleen, umbilical cord, and amniotic fluid of a stillborn puppy [14]. Dogs infected with *H. canis* usually do not show any clinical signs [11]. However, if there is a high level of parasitaemia or if the vertebrate hosts are co-infected with other vector-borne infectious agents, severe clinical manifestations could occur [15]. Severe signs of the disease can be lethargy, fever, anorexia, weight loss, lymphadenomegaly, and anaemia [16]. However, it was shown that dogs naturally infected with *H. canis* showed no clinical signs of the parasite despite its presence being confirmed by PCR tests and cytology [15]. Furthermore, eosinophilia, leukocytosis, lymphocytosis, neutrophilia, and other hematological abnormalities were present in almost all dogs analyzed [17].

At present, the prevalence of *Hepatozoon* species among Sardinian dogs is unknown. The brown dog tick, *Rhipicephalus sanguineus* sensu lato is the main vector of these pathogens [18]. Only recently, the vectorial role of other tick species belonging to the *Amblyomma, Haemaphysalis,* and *Rhipicephalus* genera has been experimentally proved, and several species from these genera are now considered potential vectors of this parasite [15,19,20,21,22]. *Rhipicephalus sanguineus* is the most widespread tick species on the island, as documented by several studies in which it emerges that these ticks are present throughout central and southern Sardinia, as well as in northern Sardinia [23]. As a result, pet dogs can potentially be exposed to *R. sanguineus* ticks and may have an increased risk of acquiring *H. canis* during the period when the ticks are most active. Therefore, the aim of this preliminary study was to determine the presence of DNA of *Hepatozoon* species in a subset of Sardinian domestic dogs.

## 2. Materials and Methods

### 2.1. Collection of Samples

The survey was conducted from March 2020 to May 2021 by collecting blood samples from dogs presented at veterinary clinics for routine medical checks. Clinics were located in different areas of Sassari and the vicinity (Figure 1). 

Each dog was subjected to a physical examination in which the body condition, attitude, capillary refill time, temperature, color of mucous membranes, and clinical signs of hepatozoonosis were determined. Each dog’s owner filled in a questionnaire that contained information about the residence locality of the dogs (province), breed, age, gender, and living environment (rural or urban). Information on the use of chemoprophylaxis practices, the history of tick infestation, and the frequency of ectoparasiticide treatment were also requested. 

Blood samples (approximately 2–3 mL) were collected by venipuncture using sterile needles, and whole blood was placed in EDTA tubes (Vacutainer sterile, R, IVD, Padova, Italy). All blood samples collected were stored at 4 °C until examination and then transported to the laboratory of the IZS Sardinia for no longer than 12 h. The illness of the dogs was also evaluated by performing a complete blood count (CBC) that was achieved using an automated hematology and chemistry analyzer (Dimension RxL Max Integrated Chemistry System (Siemens Healthcare Diagnostics, Milan, Italy)). Hematological profiles from each dog were performed and then analyzed. It consisted of red blood cell (RBC) count, hemoglobin (HGB), hematocrit (HCT), mean corpuscular volume (MCV), mean corpuscular hemoglobin (MCH), MCH concentration (MCHC), red blood cell distribution width (RDW), platelet, and white blood cell (WBC) count.

### 2.2. DNA Extraction and PCR Analysis

DNA from each sample was extracted by using the DNeasy Blood & Tissue Kit (Qiagen, Hilden, Germany) with some modifications. The incubation time was overnight, following DNA extraction according to the manufacturer’s protocols.

The gDNA extracts from dog blood samples were first screened for the presence of *Babesia*/*Theileria* spp. using previously reported primers [24], which target a 672-base pair (bp) portion of the 18S rRNA gene. The 18S rRNA target gene of *Hepatozoon* spp. was also amplified using HepF (5′-ATACATGAGCAAAATCTCAAC-3′) and HepR (5′-CTTATTATTCCATGCTGCAG-3′) as described by Inokuma et al., 2002 [25]. The total reaction volume of PCR was 25 μL that included: 9.5 μL of H2O milliQ RNasy-free, 12.5 μL of Master Mix Quantitech (Qiagen, Hilden, Germany), 1 μL of each forward and reverse primers (1 pM) and 1 μL of the DNA extracted from each sample. The amplification conditions were as follows: 15 min at 95 °C, 40 cycles of 30 s at 95 °C, 30 s of annealing temperature (60 °C), 30 s at 72 °C, and a final extension of 5 min at 72 °C. Negative and positive controls were added at each run by using ultrapure water and genomic DNA of *Hepatozoon canis* isolated from a tick, respectively. All PCR products were isolated by 1.5% (*w*/*v*) agarose gel electrophoresis in TAE buffer, stained with SYBR Safe DNA Gel Stain (Invitrogen, Carlsbad, CA, USA). DNA bands in gels were visualized using an ultraviolet transilluminator. 

A descriptive statistic (95% confidence interval, CI) was used to evaluate the infection ratio regarding sex (male or female), age, ectoparasiticide treatments, environment, provenience, and clinical signs of each dog tested (Table 1).

### 2.3. DNA Purification, Sequencing, and Phylogenetic Analyses

All amplicons were purified using a QIAquick Spin PCR Purification Kit (Qiagen, Hilden, Germany). The 18S rRNA gene was directly sequenced using the capillar sequencer ABI PRISM 3500 GENETIC ANALYZER (Applied Biosystem, Waltham, MA, USA) with the BigDye^®^ Terminator v3.1 (Sigma, St. Louis, MO, USA). DNA sequences were analyzed using the Chromas Pro software package, version 2.5.1, and their identity was confirmed by comparison to sequences available in the GenBank database using the NCBI Basic Local Alignment Search Tool (BLAST, https://blast.ncbi.nlm.nih.gov/Blast.cgi, accessed on 31 July 2023). Genetic and phylogenetic analyses were performed in all positive samples (n = 9) by PCR.

Sequences of the *Hepatozoon* 18S rRNA gene were deposited in the GenBank using the National Center for Biotechnology Information (NCBI, Bethesda, MD, USA) BankIt v3.0 submission tool (http://www3.ncbi.nlm.nih.gov/BankIt/, accessed on 31 July 2023). The phylogenetic tree was constructed using the maximum-likelihood method based on the Tamura 3-parameter model [26], identified as the best substitution model by MEGA 11 software (https://www.megasoftware.net). This option identified the best DNA/protein models by testing the dataset and then returning the most appropriate evolutionary model for analysis. Due to representing the evolutionary history of the taxa analyzed, the bootstrap consensus tree was inferred from 1000 replicates [27]. Branches corresponding to partitions reproduced in less than 50% of bootstrap replicates were collapsed. The percentage of replicate trees in which the associated taxa clustered together in the bootstrap was shown next to the branches. Initial tree(s) for the heuristic search were obtained automatically by applying Neighbor-Join and BioNJ algorithms to a matrix of pairwise distances estimated using the Tamura 3 parameter model and then selecting the topology with superior log likelihood value. To model evolutionary rate differences among sites, a discrete gamma distribution was used. The analysis involved 23 nucleotide sequences. There was a total of 584 positions in the final dataset. Evolutionary analyses were conducted in MEGA11 [28].

The sequences obtained in this study were deposited in the GenBank database (accession numbers OR350563).

## 3. Results 

### 3.1. Collection of Samples and Detection of Hepathozoon spp. by PCR

In this study, a total of fifty-one blood samples were collected from pet dogs presented at veterinary clinics located in nine districts of Sardinia, Italy. Among them, forty-two companion dogs went to the veterinary clinic for vaccination or general inspection and showed no obvious clinical signs, while nine of them needed veterinary assistance and had clinical signs such as fever, lethargy, anorexia, and lymphadenopathy. Detailed information about the sex, breed, age, clinical status, and geographic distribution of the companion dogs from this study are described in Table 1. 

From the total of analyzed blood samples, the results of PCR assay targeting the 18S rRNA showed that nine samples were positive for *Hepatozoon* infection (18%, 9/51; 95% CI: 7–29) (Figure 2), while none of the tested dogs were positive for *Babesia*/*Theileria* species.

Of the positive dogs, seven were from Sassari (78%, 7/9; 95% CI: 51–105), one from Bortigali (11%, 1/9; 95% CI: 0–31) and one from Porto Torres (11% 1/9; 95% CI: 0–31). Positive companion dogs were purebred (7/9, 73%; 95% CI: 95% CI: 51–105), female (7/9, 78%; 95% CI: 51–105), and aged from one to twelve years. Rectal temperature was elevated in three cases, while the other nine had normal temperatures. All dogs showed regular physical condition. Moreover, all positive dogs had undergone treatment for parasitic infestations. Details of each clinical sign with the percentage of the positive dogs are presented in Table 1.

In dogs with hepatozoonosis, no significant changes in hematological and biochemical values were shown. 

### 3.2. Nucleotide Sequence and Phylogenetic Analysis

In order to identify at the species level the *Hepatozoon* strains obtained in this study, all 18S rRNA-positive samples were successfully sequenced.

The results of alignments revealed that all generated 18S rRNA sequences were identical and shared the same nucleotide sequence. For this reason, only one sequence has been deposited in the GenBank database. The BLAST search of the *18S rRNA* gene sequence showed 100% nucleotide identity with *H. canis* sequences available in GenBank. The partial 18S rRNA gene sequence generated in this study was then aligned with 18S rRNA gene sequences representative of *Hepatozoon* strains identified in different hosts worldwide. The maximum likelihood phylogenetic analysis showed that the 18S sequence from this study was grouped with other 18S rRNA strains of *H. canis* (accession number: AY461378; AY731062; JF459994; KC138532; KP715301; KX818220; KU765202; MH615006) within a monophyletic-clade strongly supported that includes all previously described genotypes from different hosts species and geographic regions (Spain, Taiwan, Italy, India, Thailand, and Israel). This clade was distinct from the other *Hepatozoon* clade (Figure 3). 

## 4. Discussion

*Hepatozoon* species are parasitic hemoprotozoan organisms that infect domestic and wild animals all over the world [29]. As part of research studies and a control program on diseases transmitted by ticks, the present study analyzed blood samples collected from 51 dogs in 2020 and 2021 from different collection sites in Sardinia, Italy. To the best knowledge of the authors, the results of this study demonstrate the first detection of *H. canis* in the reference area. Being exposed to hematophagous vector bites and the related arthropod-borne pathogens, wild animal species represent a potential health hazard to domestic dog populations and humans [30]. Unlike what happens in domestic dogs, in which the disease can range from subclinical to severe symptoms that affect different organs, causing anemia and lethargy, wild animals do not develop clinical signs [31]. However, since information on wild canids is scant due to their wild nature, studies on wildlife are challenging. Currently, *H. canis* infects a wide range of carnivorous hosts worldwide, including dogs, jackals, foxes, opossums, and domestic cats. Different techniques, including indirect or direct methods, are used as diagnostic tools for *Hepatozoon* species detection. Meronts and/or monozoic cysts can be detected by histopathology in different tissues [13] as well as by indirect immunofluorescence antibody (IFA) test used for *H. canis* diagnosis in dogs with chronic infections [32]. Conventional PCR was used to detect the DNA of hemoprotozoan organisms. Among direct detection methods, molecular techniques are highly sensitive and specific for *Hepatozoon* detection as compared to other direct diagnostic tools such as blood smear procedures [10]. These last detection methods often show limited sensitivity and low specificity. In fact, although gamonts represent the final stage of the parasite’s development within the vertebrate intermediate host, they are occasionally found in blood smears. It can be related to the fact that the parasite is visible at the early stage of infection, chronic aparasitemic infection, or low parasitemia [32].

In this study, although the number of analyzed samples was rather small, the prevalence of *H. canis* infection recorded (9/51; 18%) was high and indicated that *H. canis* occurs on the island. However, as infected dogs frequently show unspecific symptoms, estimating the real burden of the disease is a challenge, and it could lead to a probable significant lack of prevention of long-term complications for at-risk dogs. 

Unfortunately, only a few studies have been performed in Italy on canine hepatozoonosis, and a comparison in terms of infection rates among the different regions of Italy cannot be conducted. However, Ebani et al., 2015 [33] have reported *H. canis* presence in hunting dogs from central Italy (32.5%). In addition, a molecular prevalence of 3.63% was reported by Cassini et al. (2009) [34] in central–northern Italy from farm and hunting dogs. 

This rate variation is likely attributed to several factors, including the number of sample sizes as well as the presence and spread of tick infestations in the different areas of Italy [35]. In fact, the occurrence of *H. canis* in dogs is due to the presence of the competent vector *Rhipicephalus sanguineus* sensu lato [18] and to the spread of this pathogen in the dog population [36,37,38]. *R. sanguineus* s.l. is the most widespread tick species in Sardinia, as the favorable climatic conditions on the island allow the development and survival of these tick species throughout the year [39]. Moreover, an inferential statistic was not performed in this study since the low number of samples did not allow us to perform inference on variables, so we just used a descriptive statistic. 

In this study, although *H. canis* infection was confirmed by PCR in nine dogs (18%; 95% CI: 8–28), only six (12%; 95% CI: 12–76) showed clinical unspecific symptoms; the others did not show any clinical signs. This fact could reflect the presence of subclinical or mild infections that, as reported above, are a challenge for the diagnosis. In fact, in the majority of reported cases, the diagnosis is made when complications occur, and clinical signs are evident only if the parasitemic load is high or if the *H. canis* coinfection with other vector-borne infectious agents is detected in the same host [15,19]. The lack of hematological alterations could also depend on the interaction that occurs between the host and the parasite. Indeed, since the dog may be able to deal with the parasite, both the parasitic burden and the low pathogenicity of *H. canis* should also be considered. In this study, symptomatic dogs presented fever and muscle pain, and it was in accordance with the main complications caused by *H. canis* infection that involves fever, paralysis, anorexia, wasting, anemia, eye discharge, and weakness of the hind legs [19]. 

Our study reveals that the *Hepatozoon* prevalence was higher in females (23%; 95% CI: 8–38) than in male dogs (10%; 95% CI: 0–23). These results were in contrast with previous reports in which no sex predisposition towards hepatozoonosis was highlighted in infected dogs [40]. However, we think that these differences may be due to the different environments that would expose the dogs to more ticks. 

With regard to age, the prevalence of *H. canis* infection from this study was lower in dogs that ranged from 5 to 10 years (14%; 95% CI: 0–32) in comparison to dogs that had an age between 1 and 5 years (20%; 95% CI: 4–36) and to those that were above ten years (29%; 95% CI: 0–63). Since the number of samples here analyzed was rather low and just a descriptive statistical analysis was conducted, we cannot compare these results with other reports in which a large number of samples were involved. Nevertheless, there is discordant data in the literature concerning the prevalence of *H. canis* in the canine population. From previous studies, it emerged that the rate of *H. canis* infection was not significantly associated with age in the dog population [29,40,41]. On the contrary, other investigations highlighted a higher infection prevalence in adult dogs in comparison with the younger ones [42,43]. 

Although several studies report that mild non-regenerative anemia, leukocytosis with neutrophilia and monocytosis, or mild thrombocytopenia [44,45] are the most common laboratory findings after *Hepatozoon canis* infection, our data demonstrated that no hematological abnormalities were detected in any of the nine positive dogs. We can suppose that hematological changes are less pronounced in hepatozoonosis and abnormalities are relievable when co-infection with one or more vector-borne diseases occurs. We found no cases of *Babesia* infection in dogs analyzed in this study. However, several studies revealed the presence of co-infections with *Babesia* species in dogs, as well as in ticks that harbor the DNA of more than one pathogen. The occurrence of co-infections with *Babesia* and *Hepatozoon* could increase the pathogenicity in the infected dogs, causing more complications that may occur in serious cases.

In this study, positive dogs were all from Sassari and the surrounding area. We demonstrate that this district is not free of hepatozoonosis, and these findings indicate the need for future studies to address the study of ticks from this region where this protozoan has never been described thus far. Moreover, since one of the limitations of the current work is based on the lack of data on the blood smear, this direct test, as well as other direct diagnostic methods, should be conducted to include asymptomatic cases. 

Although most of the dogs had received antiparasitic treatment, which protects the dogs and limits the circulation of pathogens, Sardinian dogs were still highly exposed to vector-borne hepatozoonosis. Numerous studies attest that a greater number of dogs, not only stray or shelter dogs but also the owned ones, do not receive adequate preventive treatments to protect the animals from arthropods (e.g., ticks, fleas, sandflies, or mosquito infestations). It increases the risk of ectoparasite infestations, transmission of diseases, and the presence of infected arthropods in domestic environments. Furthermore, due to the lack of effective treatment, therapy for dogs infected with *Hepatozoon* remains a major problem [46]. In fact, hepatozoonosis is generally considered a lifelong infection in dogs, and no known treatment regimen completely eliminates the pathogen from animals.

## 5. Conclusions

The resulting data, although still preliminary, are interesting since they provide the first valuable information on the presence of *H. canis* infection in dogs from Sardinia, Italy. Considering that the molecular characterization of this pathogen has never been carried out on the island, we hope that similar studies will be conducted in this area. Therefore, improving knowledge about hepatozoonosis will involve the establishment of control programs, which will be fundamental to adequately manage and control the spread of the disease. The development and implementation of specific differential diagnostic tests will be an essential prerequisite for establishing the highest development of a diagnostic approach for *Hepatozoon* sp. detection.

## Figures and Tables

**Figure 1 vetsci-10-00640-f001:**
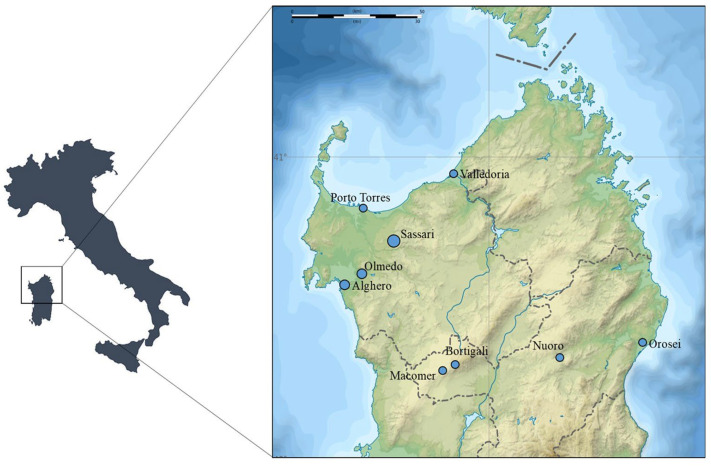
Map of Sardinia indicating the locations (Valledoria (40°55′ N; 8°49′ E), Porto Torres (40°50′ N; 8°24′ E), Sassari (40°43′ N; 8°33′ E), Olmedo (40°39′ N; 8°22′ E), Alghero (40°33′ N; 8°19′ E), Bortigali (40°17′ N; 8°50′ E), Macomer (40°15′ N; 8°46′ E), Nuoro (40°19′ N; 9°18′ E), and Orosei (40°22′ N; 9°41′ E)) of the veterinary clinics from which dogs have been screened for *Hepatozoon* and *Babesia/Theileria* species.

**Figure 2 vetsci-10-00640-f002:**
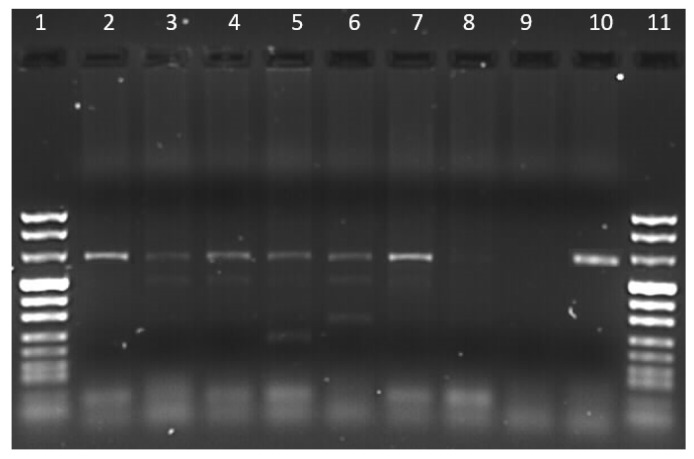
Agarose gel electrophoresis (1.5%) of PCR products of *Hepatozoon* spp. Lanes 1 and 11: size marker; lanes 2–7: positive samples; lane 8: negative sample; lane 9: K−; lane 10: K+.

**Figure 3 vetsci-10-00640-f003:**
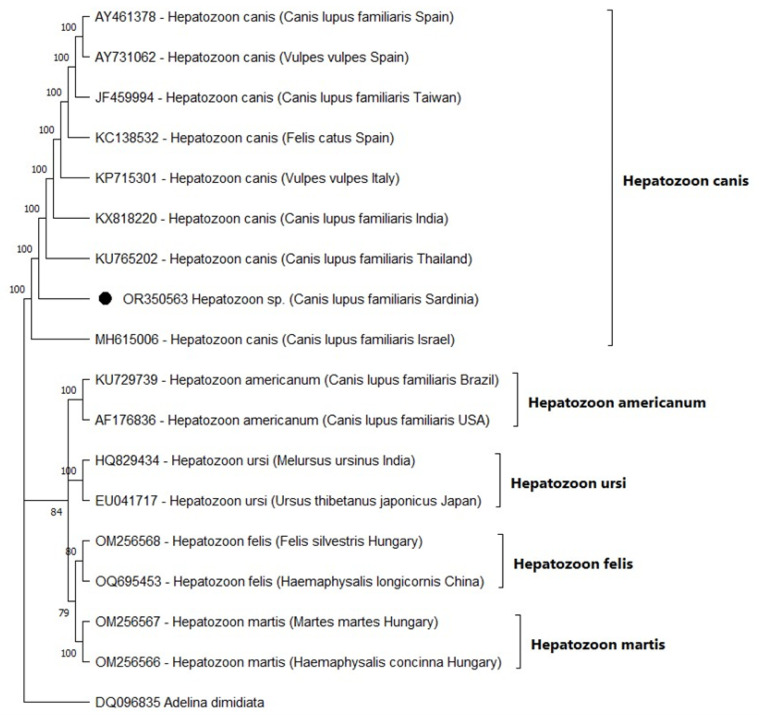
Phylogenetic tree presenting the relationships of 18S rRNA gene sequences from *Hepatozoon* isolates from this study and the other genotypes deposited in GenBank. The accession numbers of each strain are presented in the tree. The sequence of *Adelina dimidiata* served as an outgroup. Numbers at the nodes indicate bootstrap support from 1000 iterations.

**Table 1 vetsci-10-00640-t001:** Variables of 51 dogs tested for *Hepatozoon* spp. in Sardinia analyzed by sex, age, tick infestation, environment, location, and clinical signs. 95% CI.

Parameters	Variables	Sample Analyzed	Positive Dogs (%)	95% CI
Gender	Male	20	2 (10)	0–23
	Female	31	7 (22.6)	8–38
Age (years)	<1	2	0	0
	1–5	25	5 (20)	4–36
	5–10	14	2 (14.3)	0–32
	>10	7	2 (28.6)	0–63
	Not determined	3	0	0
Ectoparasiticide treatments	Yes	46	8 (17.4)	6–28
	No	5	1 (20)	0–55
Environment	Urban	19	6 (32)	11–43
	Rural	32	3 (9.4)	0–19
Geographical location	Sassari	48	9 (18.8)	8–30
	Nuoro	3	0	0
Clinical signs	Lymphadenomegaly	7	1 (14)	0–40
	Fever	3	1 (33)	0–86
	Anorexia	2	0	0
	Muscular pain	11	4 (36)	8–64
	No clinical symptoms	28	3 (11)	0–23

## Data Availability

Not applicable.

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
