# Peer review of "Molecular Survey of Hepatozoon canis Infection in Domestic Dogs from Sardinia, Italy"

_vetsci, 2023, doi:10.3390/vetsci10110640_

Round 1

Reviewer 1 Report

Comments and Suggestions for Authors

The paper written by Chisu et al. investigated 51 owned dogs from different areas of Sardinia Island for Piroplasms (i.e., Hepatozoon, Babesia and Theileria) by PCR.

Information concerning sampled dogs (i.e., age, location, clinical signs, treatment against ectoparasite, and environment) were collected during the clinical health checks. 

Nine dogs were positive for H. canis, while none of them showed positivity for Babesia and Theileria. In addition, the authors reported that 6/9 positive dogs had clinical signs referable to H. canis infection.

Literature concerning Piroplasm in Sardinia is scant. Although the present study is preliminary (only 51 sampled dogs), information on this topic could be useful to implement knowledge on spread and distribution of these parasites in the Mediterranean area.  

The paper has been well prepared but need improvement in the following aspects. In addition, I suggest checking more accurately English.

Introduction

1.     Line 39: “the host cells and is composed…” more likely “the host cells and IT is composed…”

2.     Line 44: “Hematophagous invertebrate” needs plural “invertebrates”

3.     Line 49: “however, two species infect mammal leukocytes….”  more likely “two species infectING mammal leukocytes”

4.     Line 61: “gamonte stage” without E at gamont end

5.     Lines 71 and 72: I suggest modifying the sentence to “...for H. canis, CONVERSALY the vertical transmission has been REPORTED” to avoid repetition

6.     Line 73: As you said the intrauterine transmission has been demonstrated, indeed you reported two references n. 11 (Schäfer et al.) and n. 13 (Murata et al.) about that. Nevertheless, you cited only the Japanese study about a positive peripheral blood smear. I suggest adding even the recent study by Schäfer in which spleen, umbilical cord and amniotic fluid of a stillborn puppy were reported positive to H. canis. DOI https://doi.org/10.1186/s13071-022-05392-7

Materials and methods

7.     Line 105: I suggest modifying “…were also obtained” to “…were also collected” to avoid repetition.

8.     I suggest adding a sentence about the statistical analysis. If you did not perform it explain why. If you did, please state the kind of test used. I could suppose that the low number of samples did not allow to perform inference on variables, so you just used a descriptive statistic. However, you should state. Moreover, to calculate the CI you should have an idea about the samples size. How did you calculate it? Which is your expected prevalence? Is it based on similar studies? Please add in the text.

Results

9.     Line 178: “42 of the companion dogs….” more likely “42 companion dogs”.

10.     Lines 187 and 188: The sentence “none of the owners suspected VBD in their pets and none of the dogs tested positive by 18S rRNA PCR for Babesia/Theileria species” is not clear. I don’t understand the link between the information. Stating that the owners do not suspect VBDs is not scientifically objective. What kind of information give it to the reader? Owners don’t recognize the clinical signs of VBDs? I suppose it is true, they don’t. However, it is not a justification for the fact that no dog tested positive for Babesia/Theileria. I recommend deleting “none of the owners suspected VBD in their pets” and rephrase.

11.   In line 199, you reported “no significant changes in haematological […] values were shown”. I suppose that to state presence of anaemia in dog a CBC test should be done. Was a blood smear included in that test?  You should add it.

Discussion

12.    Lines 228-230: Information on wild canids are scant due to their wild nature, indeed, studies on wildlife in general are challenging. In my opinion, this sentence is too speculative. Add a reference or modify the sentence please please.

13.    Line 239: linking to comment 11, if a blood smear has been done, here it may be added that low sensitivity is confirmed by your negative CBC in molecularly H.canis-positive dogs. If not, you should state that this a limitation of your study.

14.   In line 263 you state that “only 4 dogs showed clinical unspecific symptoms”. According toTable 1, positive dogs with clinical signs are 6. I suppose there is a mistake in the reported data in the text or in the table. Please modify.

15.     Line 269: I suggest changing “the MAIN complication caused by H. canis infection” to “the MOST FREQUENT complication” since, as you say some lines above, the clinical signs are several and unspecific.

16.     In lines 285-294, you didn’t consider that the lack of hematological alterations could also depend on the interaction that occurs between host and parasite. Both the parasitic burden and the low pathogenicity of H. canis must be considered, indeed, the dog may be able to deal with the parasite. In my opinion, you consider only the absence of co-infestation with Babesia spp., and this is not sufficient, you should rephrase and add these possibilities.

17.     Reference number 39 is missing in the text.

Comments on the Quality of English Language

I suggest checking more accurately English.

Author Response

Introduction

  1. Line 39: “the host cells and is composed…” more likely “the host cells and IT is composed…”

Response: Done

  1. Line 44: “Hematophagous invertebrate” needs plural “invertebrates”

Response: Done

  1. Line 49: “however, two species infect mammal leukocytes….”  more likely “two species infectING mammal leukocytes”

Response: Corrected

  1. Line 61: “gamonte stage” without E at gamont end

Response: It has been modified

  1. Lines 71 and 72: I suggest modifying the sentence to “...for H. canis, CONVERSALY the vertical transmission has been REPORTED” to avoid repetition

Response: Corrected

  1. Line 73: As you said the intrauterine transmission has been demonstrated, indeed you reported two references n. 11 (Schäfer et al.) and n. 13 (Murata et al.) about that. Nevertheless, you cited only the Japanese study about a positive peripheral blood smear. I suggest adding even the recent study by Schäfer in which spleen, umbilical cord and amniotic fluid of a stillborn puppy were reported positive to H. canis.DOI https://doi.org/10.1186/s13071-022-05392-7

Response: Added

Materials and methods

  1. Line 105: I suggest modifying “…were also obtained” to “…were also collected” to avoid repetition.

Response: Modified

  1. I suggest adding a sentence about the statistical analysis. If you did not perform it explain why. If you did, please state the kind of test used. I could suppose that the low number of samples did not allow to perform inference on variables, so you just used a descriptive statistic. However, you should state. Moreover, to calculate the CI you should have an idea about the samples size. How did you calculate it? Which is your expected prevalence? Is it based on similar studies? Please add in the text.

Response: Thank you for your suggestion.  It has been added to the M&M section that now reads:

“The statistical analysis was not performed in this study since the low number of samples did not allow to perform inference on variables, so we just used a descriptive statistic. With this scope the 95% confidence interval (CI) was used to evaluate the infection ratio regarding sex (male or female), age, ectoparasiticide treatments, environment, provenience and clinical signs of each dog tested (Table 1).”

Results

  1. Line 178: “42 of the companion dogs….” more likely “42 companion dogs”.

Response: Corrected

  1. Lines 187 and 188: The sentence “none of the owners suspected VBD in their pets and none of the dogs tested positive by 18S rRNA PCR for Babesia/Theileria species” is not clear. I don’t understand the link between the information. Stating that the owners do not suspect VBDs is not scientifically objective. What kind of information give it to the reader? Owners don’t recognize the clinical signs of VBDs? I suppose it is true, they don’t. However, it is not a justification for the fact that no dog tested positive for Babesia/Theileria. I recommend deleting “none of the owners suspected VBD in their pets” and rephrase.

Response: Deleted

  1. In line 199, you reported “no significant changes in haematological […] values were shown”. I suppose that to state presence of anaemia in dog a CBC test should be done. Was a blood smear included in that test?  You should add it.

Response: The blood smear was not performed, and as highlighted in the discussion, it was one of the limitations of this study.

Discussion

  1.   Lines 228-230: Information on wild canids are scant due to their wild nature, indeed, studies on wildlife in general are challenging. In my opinion, this sentence is too speculative. Add a reference or modify the sentence please please.

Response: The sentence has been modified and a new reference has been added

  1.   Line 239: linking to comment 11, if a blood smear has been done, here it may be added that low sensitivity is confirmed by your negative CBC in molecularly H.canis-positive dogs. If not, you should state that this a limitation of your study.

Response: Added

  1.  In line 263 you state that “only 4 dogs showed clinical unspecific symptoms”. According to Table 1, positive dogs with clinical signs are 6. I suppose there is a mistake in the reported data in the text or in the table. Please modify.

Response: Indeed, it was a mistake in the text and it has been corrected.

  1. Line 269: I suggest changing “the MAIN complication caused by H. canis infection” to “the MOST FREQUENT complication” since, as you say some lines above, the clinical signs are several and unspecific.

Response: Corrected

  1. In lines 285-294, you didn’t consider that the lack of hematological alterations could also depend on the interaction that occurs between host and parasite. Both the parasitic burden and the low pathogenicity of H. canis must be considered, indeed, the dog may be able to deal with the parasite. In my opinion, you consider only the absence of co-infestation with Babesia spp., and this is not sufficient, you should rephrase and add these possibilities.

Response: Thank you for your suggestion. This hypothesis has been added in the discussion.

  1.    Reference number 39 is missing in the text.

Response: Added

Reviewer 2 Report

Comments and Suggestions for Authors

1. A map should be added to show the sampling sites.

2. line 18, genes should be gene.

3. line 22, detected here.

4. line 31, increasing.

5. line 49, acarines should be ticks.

6. line 79, showed no clinical signs, however, dogs naturally infected with H. canis showed clinical signs in other study

7. line 88, R. sanguineus, "R." should be given the full name in the first of sentence.

8. line 98, "for routine medical checks", no dog present clinical signs?

9. lines 128-131 and 133-135, two PCR reaction systems?

10. μl, should be consistent.

11. line 143, genes should be gene.

12. line 155-157 should be deleted.

13. line 153, nine positive samples, why only one accession number?

14. line 167, Tamura 3 parameter model, why was this model used?

15. line 172, "1st+2nd+3rd+Noncoding", why? 18S gene was not translated into protein.

16. line 187, Table 2 should be table 1.

17. lines 228-230, "Unlike what happens in domestic dogs (in which the disease can affect different organs causing anemia and lethargy), wild canines do not develop clinical signs.", add references.

18. line 279, "younger dogs" and "mature ones", how to define? delete lines 280-283. 

19. line 279 "the prevalence of H. canis infection from this study was higher in young (78%; 95% CI: 51-105) than adult dogs (22%; 95% CI: 11-33) " , which may be influenced by many factors, not "susceptible".

Comments on the Quality of English Language

English language only needs minor editing.

Author Response

  1. A map should be added to show the sampling sites.

Response: Added

  1. line 18, genes should be gene.

Response: Modified

  1. line 22, detected here.

Response: Modified

  1. line 31, increasing.

Response: Modified

  1. line 49, acarines should be ticks.

Response: Modified

  1. line 79, showed no clinical signs, however, dogs naturally infected with H. canis showed clinical signs in other study

Response: Modified

  1. line 88, R. sanguineus, "R." should be given the full name in the first of sentence.

Response: Modified

  1. line 98, "for routine medical checks", no dog present clinical signs?

Response: Modified

  1. lines 128-131 and 133-135, two PCR reaction systems?

Response: Modified

  1. μl, should be consistent.

Response: Modified

  1. line 143, genes should be gene.

Response: Modified

  1. line 155-157 should be deleted.

Response: Deleted

  1. line 153, nine positive samples, why only one accession number?

Response: This explanation has now been added in the text that reads:

“The results of alignments revealed that all generated 18S rRNA sequences were identical and shared the same nucleotide sequence. For this reason, only one sequence has been deposited in the GenBank database.”

  1. line 167, Tamura 3 parameter model, why was this model used?

Response: The sentence has been modified as follows:

This option identified the best DNA/protein models by testing the dataset and then returning the most appropriate evolutionary model for analysis.”

  1. line 172, "1st+2nd+3rd+Noncoding", why? 18S gene was not translated into protein.

Response: Modified

  1. line 187, Table 2 should be table 1.

Response: Modified

  1. lines 228-230, "Unlike what happens in domestic dogs (in which the disease can affect different organs causing anemia and lethargy), wild canines do not develop clinical signs.", add references.

Response: Added

  1. line 279, "younger dogs" and "mature ones", how to define? delete lines 280-283. 

Response: Deleted

  1. line 279 "the prevalence of H. canis infection from this study was higher in young (78%; 95% CI: 51-105) than adult dogs (22%; 95% CI: 11-33) " , which may be influenced by many factors, not "susceptible".

Response: Modified

Comments on the Quality of English Language

English language only needs minor editing.

Response: The English has been edited

Reviewer 3 Report

Comments and Suggestions for Authors

Dear authors, I just have few minor comments in the PDF version in comments. Very nice, short paper with important information for veterinarians

Author Response

Please, add the order of Hepatozoon as it was mentioned before

Response: Added

Is it biochemistry or biology and if it is biochemistry why it is important for this study?

Response: Corrected

serving instead of utilize

Response: Modified

please confirm that gametogonic and gamegony (row 55) are different terms

Response: Modified sentence

Just do not begin the sentence with a number

Response:Changed

should be 39.2

Response: Modified